# ClpG Provides Increased Heat Resistance by Acting as Superior Disaggregase

**DOI:** 10.3390/biom9120815

**Published:** 2019-12-02

**Authors:** Panagiotis Katikaridis, Lena Meins, Shady Mansour Kamal, Ute Römling, Axel Mogk

**Affiliations:** 1Center for Molecular Biology of the University of Heidelberg (ZMBH) and German Cancer Research Center (DKFZ), DKFZ-ZMBH Alliance, 69120 Heidelberg, Germany; p.katikaridis@zmbh.uni-heidelberg.de (P.K.); lena.meins@mail.de (L.M.); 2Department of Microbiology, Tumor and Cell Biology, Karolinska Institute, 17177 Stockholm, Sweden; shady.kamal@ki.se (S.M.K.); ute.romling@ki.se (U.R.)

**Keywords:** protein disaggregation, heat resistance, chaperone, protein aggregation, AAA protein, Hsp100

## Abstract

Elevation of temperature within and above the physiological limit causes the unfolding and aggregation of cellular proteins, which can ultimately lead to cell death. Bacteria are therefore equipped with Hsp100 disaggregation machines that revert the aggregation process and reactivate proteins otherwise lost by aggregation. In Gram-negative bacteria, two disaggregation systems have been described: the widespread ClpB disaggregase, which requires cooperation with an Hsp70 chaperone, and the standalone ClpG disaggregase. ClpG co-exists with ClpB in selected bacteria and provides superior heat resistance. Here, we compared the activities of both disaggregases towards diverse model substrates aggregated in vitro and in vivo at different temperatures. We show that ClpG exhibits robust activity towards all disordered aggregates, whereas ClpB acts poorly on the protein aggregates formed at very high temperatures. Extreme temperatures are expected not only to cause extended protein unfolding, but also to result in an accelerated formation of protein aggregates with potentially altered chemical and physical parameters, including increased stability. We show that ClpG exerts higher threading forces as compared to ClpB, likely enabling ClpG to process “tight” aggregates formed during severe heat stress. This defines ClpG as a more powerful disaggregase and mechanistically explains how ClpG provides increased heat resistance.

## 1. Introduction

A sudden increase in temperature represents a threat to all organisms, unicellular organisms in particular, including bacteria. Severe heat stress affects the structural integrity of proteins to cause their misfolding ultimately resulting in protein aggregation. A wide variety of cellular proteins, including essential proteins that act, e.g., in transcription, translation, and cell division, can readily form protein aggregates [1,2]. The heat-induced loss of protein functionality by aggregation consequently causes loss of cellular viability. A reduction of the number of contaminating microorganisms by heat treatment is therefore a classical procedure in food production processes and in the sterilization of medical equipment.

To resist severe heat stress bacteria employ diverse protective measures including alteration in membrane composition and fluidity, accumulation of compatible solutes and induction of the heat shock response [3]. The increased production of specific chaperones, termed disaggregases, enables cells to revert protein aggregation and to restore functionality of otherwise lost proteins. The most widespread bacterial disaggregation machinery is composed of an Hsp70 chaperone system (DnaK in bacteria) that cooperates with a cognate Hsp100 disaggregase (ClpB in bacteria) [4]. Hsp100 disaggregases are members of the AAA+ protein superfamily. They harbor two conserved AAA domains that mediate hexamer formation and fuel disaggregation by hydrolyzing ATP. The derived energy is used to extract aggregated proteins via a threading mechanism that is executed by pore-located aromatic residues. The positions of these residues constantly change during the ATPase cycle of AAA proteins dragging a substrate through the central translocation channel [5,6].

The ClpB/DnaK bi-chaperone system provides heat resistance to bacteria but also to fungi and plants [7,8,9]. ClpB is present in all Gram-negative and most Gram-positive bacteria. Only distinct bacterial species such as *Bacillus subtilis* do not encode for this disaggregase [10]. Notably, the disaggregation activity of ClpB essentially requires cooperation with DnaK [11,12,13]. DnaK binds first to the surface of protein aggregates with the help of its cochaperone DnaJ and recruits ClpB in a subsequent step [14,15]. This recruitment is coupled to ClpB activation, causing an increase in ATPase and threading activities [16,17,18]. DnaK-mediated activation involves the ClpB M-domains, which form a coiled-coil structure engulfing the first ATPase ring [19]. Binding of DnaK to M-domains keeps them apart from the ring, leading to activation of ClpB. ClpB mutants that cause permanent dissociation of M-domains result in persistent cytotoxic activation of the protein [16,20]. Accordingly, ClpB activation by DnaK is only transient and ClpB exhibits reduced unfolding power as compared to other Hsp100 proteins, even when engaged in protein disaggregation [21].

A particular attention has recently been drawn to ClpG (ClpK), which represents a novel family of Hsp100 disaggregases in bacteria. The standalone ClpG directly binds to aggregated proteins and does not require cooperation with DnaK [22]. ClpG (ClpK) was first described in a persistent nosocomial *Klebsiella pneumoniae* strains and proposed to be the molecular basis of increased survival upon severe heat stress [23]. ClpG has subsequently been identified in various Gram-negative bacteria including pathogenic species like *Enterobacter sp*, *Pseudomonas aeruginosa*, *Chronobacter sp*, *Salmonella enterica*, and *Escherichia coli* [24,25,26,27]. ClpG belongs to a horizontally transferred gene cluster termed LHR (locus of heat resistance) or TLPQC (transmissible locus for protein quality control) that comprises up to 16 open reading frames [25,26]. This gene cluster is either present on mobile genomic islands or located on conjugative plasmids, and thus can be laterally transferred to other bacteria [3]. ClpG is a conserved component of this locus and represents an unequivocal predictor of increased bacterial heat resistance. Of note, a ClpG homolog is also encoded on the core genome of *P. aeruginosa*, and exhibits biochemical characteristics of ClpG-family members such as standalone disaggregation and high ATPase activity [22]. In world-wide distributed *P. aeruginosa* clone C, ClpG_GI_ encoded on a genomic island acts additively with chromosomally encoded ClpG [22].

Superior heat resistance provided by ClpG has a substantial impact on bacterial survival during exposure to elevated temperatures up to 70 °C. Those conditions occur upon mild thermal sterilization of specific medical equipment such as endoscopes, but also during food production and processing. A strong increase of the prevalence of *clpG* bearing *E. coli* strains isolated from raw milk cheese and steam-pasteurized beef, as compared to the total number of *E. coli* genomes indicates the contribution of *clpG* and the LHR [24,28] and strongly suggests that bacteria harboring *clpG* have an advantage during temporal exposure to lethal temperatures. Along the same line *clpG* might contribute to bacterial persistence in the hospital setting and has been associated with the emergence of multidrug resistant *K. pneumoniae* clones and extended spectrum beta-lacatamase (ESBL) *E. coli* [23,29,30].

The expression of *clpG* can increase heat resistance in various bacterial species such as *E. coli*, *K. pneumoniae*, and *P. aeruginosa* [22,31]. However, it remains unclear whether the beneficial effect on temperature tolerance stems uniquely from intrinsic ClpG features or whether the presence of the two independent disaggregation systems ClpB/DnaK and ClpG, increase the overall disaggregation power. A systematic comparison of the disaggregation activities of the “classical” ClpB/DnaK bi-chaperone disaggregase and novel, standalone ClpG has not been performed yet. Characterization of the disaggregation potential of ClpG will aid the understanding of its specific contribution to bacterial heat resistance.

Here, we show that ClpG exhibits superior disaggregation activity as compared to ClpB/DnaK towards a variety of model substrates in vitro and in vivo. The differences between both disaggregation systems are most pronounced when protein aggregates were formed under conditions of extreme heat stress. Those ‘extreme’ aggregates were almost resistant towards ClpB-dependent disaggregation, while ClpG robustly processed such denatured proteins. We can link this robust ClpG activity to a particularly high unfolding power, enabling ClpG to process “tight” aggregates that form at high temperatures. These findings hint to how ClpG can protect bacterial cells from severe heat treatments.

## 2. Materials and Methods

### 2.1. Strains, Proteins and Plasmids

*E. coli* strains used were derivatives of MC4100. *E. coli* ClpB and ClpB-K476C were purified after overproduction from *E. coli ΔclpB::kan* cells using pDS56-derived expression vectors [16]. *P. aeruginosa* ClpG_GI_ (name ClpG throughout this manuscript) and ClpG_GI_-E383A/E732A were purified after overproduction in *E. coli* BL21 cells using pET24a-derived expression vectors. ClpB and ClpG mutant derivatives were generated by Quickchange one step site directed mutagenesis. PCR amplicons were incubated with DpnI, which only digests methylated DNA, for 1 h at 37 °C and subsequently transformed into *E. coli* XL1 blue cells. All mutations were verified by DNA sequencing. All proteins harbor a C-terminal His_6_-tag and were purified using Ni-IDA (Macherey-Nagel) following the instructions provided by the manufacturer. Subsequently, pooled protein fractions were subjected to size exclusion chromatography (Superdex S200, GE Healthcare, Freiburg, Germany) in buffer A1 (50 mM Tris pH 7.5, 50 mM KCl, 10 mM MgCl_2_, 5% (*v*/*v*) glycerol, 2 mM DTT) for ClpB and buffer A2 (50 mM Tris pH 7.5, 150 mM KCl, 5% (*v*/*v*) glycerol, 2 mM DTT) for ClpG.

Purifications of DnaK, DnaJ, GrpE, Firefly Luciferase and Luciferase-YFP were performed as described previously [16,21,32]. Pyruvate kinase of rabbit muscle, Malate Dehydrogenase of pig heart muscle, Citrate Synthase from porcine heart and α-Glucosidase from *Saccharomyces cerevisiae* were purchased from Sigma. Protein concentrations were determined with the Bradford assay (Biorad).

### 2.2. Biochemical Assays

#### 2.2.1. In Vitro Disaggregation Assays

Disaggregation activity was determined by following the disaggregation of diverse model substrates in buffer A (50 mM Tris pH 7.5, 50 mM KCl, 10 mM MgCl_2_, 2 mM DTT): heat-aggregated Malate Dehydrogenase (2 μM, 30 min at 47 °C), Citrate Synthase (2 μM, 30 min at 45 °C), α-glucosidase (2 μM, 30 min at 47 °C) and Luciferase (25 or 200 nM, 15 min at 42–53 °C). Chaperones were used at the following concentrations: 1 μM ClpB (wild type or K476C) or ClpG, *E. coli* Hsp70 system, KJE: 1 μM DnaK, 0.2 μM DnaJ, 0.1 μM GrpE. Disaggregation reactions were performed in buffer A containing an ATP regenerating system (2 mM ATP, 3 mM phosphoenolpyruvate, 20 ng/μL Pyruvate Kinase). Refolding of aggregated Luciferase was monitored by determining Luciferase activities using a Lumat LB 9507 (Berthold Technologies, Bad Wildbad, Germany). The activity of native Luciferase was set as 100%. For other substrates disaggregation was monitored by turbidity measurement at an excitation and emission wavelength of 600 nm (PerkinElmer LS50B spectrofluorimeter, Hamburg, Germany). Disaggregation rates were calculated from the linear decrease in aggregate turbidity.

#### 2.2.2. Ex Vivo Disaggregation of Proteins in Heat-Denatured Crude Extracts

Disaggregation of bulk proteins isolated from heat denatured crude extracts of *E. coli* Δ*clpB* cells expressing Luciferase was performed as described [1] with minor modifications. Bacterial cells were cultured at 30 °C in LB medium until mid-logarithmic growth phase. Cells were collected by centrifugation, resuspended in buffer C (50 mM HEPES pH 7.5, 50 mM KCl, 10 mM MgCl_2_, 2 mM DTT) and mechanically lysed by using a French press cell. The cell lysate was cleared by centrifugation (15,000 rpm, 30 min, 4 °C) and dialyzed against buffer B overnight. Dialysed cell lysates were centrifuged again (15 min, 13,000 rpm, 4 °C) and aliquots were prepared. A fraction of the cell lysate was labeled with *N*-Succinimidyl [2–^3^H] propionate (PerkinElmer, NET632H005MC) for 2 h at room temperature. Non-reacted labeling reagent was removed by dialysis against buffer C. To induce the formation of protein aggregates ex vivo, 100 μL cell lysate (0.5–1 μg/μL) was incubated for 15 min at 46 °C and aggregates were isolated by centrifugation (13,000 rpm, 15 min, 4 °C). Aggregated proteins were resuspended in buffer B and incubated with disaggregases (1 μM ClpB/ClpB-K476C + KJE: 1 μM DnaK, 0.2 μM DnaJ, 0.1 μM GrpE, 1 μM ClpG_GI_) at 30 °C in the presence of an ATP regenerating system. Luciferase activities were determined in non-labeled cell lysates prior heat shock (set as 100%) and during disaggregation reactions using a Lumat LB 9507 (Berthold Technologies). 2 μL of an ongoing disaggregation reaction were mixed with 125 μL Luciferase assay buffer (25 mM glycylglycine pH 7.4, 12.5 mM MgSO_4_, 5 mM ATP). 125 μL 0.25 mM Luciferin were subsequently injected and luminescence was determined for 2–5 s. Disaggregation of labeled cell lysates was monitored by determining the amount of re-solubilized ^3^H-labeled aggregated proteins. Solubilized proteins were separated from still aggregated ones by centrifugation (13,000 rpm, 15 min, 4 °C) and quantified by scintillation counting (Beckman LS 6000IC).

#### 2.2.3. Monitoring YFP Unfolding During Protein Disaggregation

In buffer A, 200 nM Luciferase-YFP was denatured at 46 °C for 15 min. Sample turbidity (excitation and emission wavelengths: 600 nm) and YFP fluorescence (excitation: 505 nm, emission: 525 nm) were simultaneously recorded in the absence and presence of disaggregases using a PerkinElmer LS50B spectrofluorimeter.

### 2.3. Characterization of Luciferase Aggregates

#### 2.3.1. Dynamic Light Scattering Measurements

In buffer A, 100 μL of 25 or 200 nM Luciferase was denatured at 42–53 °C for 15 min. Protein aggregates were transferred to a quartz cuvette and light scattering was determined using a Dyn-aPro-MS/X instrument (Protein Solutions Ltd., High Wycombe, Bucks, UK). Particle sizes were calculated by the software DYNAMICS (version 6.0) supplied by the manufacturer.

#### 2.3.2. Bis-ANS Binding

For Bis-ANS binding, 5 μM Bis-ANS (4,4′-Dianilino-1,1′-binaphthyl-5,5′-disulfonic acid dipotassium salt, Thermofisher) was added to 25 nM or 200 nM aggregated Luciferase and incubated for 10 min at room temperature. Bis-ANS binding was determined by recording an emission spectrum using a PerkinElmer LS50B spectrofluorimeter (390 nm excitation, 430–600 nm emission scan).

### 2.4. Aggregate Binding Assay

In buffer A 1.5 μM ClpG_GI_-E383A/E732A (ClpG-DWB) was incubated with aggregated luciferase (2 μM, generated by incubation at 46 °C for 15 min) in the presence of 2 mM ATP at 25 °C for 10 min. Then, 1 μM DnaK together with 0.2 μM DnaJ were added to selected reactions. Soluble and insoluble fractions were separated by centrifugation at 13,000 rpm for 25 min at 4 °C. The pellet fraction was washed once with 300 μL buffer A and centrifuged again at 13,000 rpm for 10 min at 4 °C. Binding assays were performed in low binding micro tubes (Sarstedt). Supernatant and pellet fractions were mixed with protein sample buffer and analyzed by Coomassie staining after SDS-PAGE (8–16% polyacrylamide gradient gels). Binding assays were repeated three times independently. As control purified ClpG-DWB without aggregated Luciferase was subjected to the same protocol. ClpG-DWB band intensities in soluble (I_s_) and insoluble (I_i_) fractions were quantified using ImageJ and the percentage of ClpG-DWB present in the pellet fraction was determined by calculating I_i_/(I_s_ + I_i_).

### 2.5. In Vivo Disaggregation Assays

#### 2.5.1. Luciferase Reactivation

*E. coli* Δ*clpB* cells harboring the plasmid placIq-Luciferase (for constitutive expression of Luciferase) and pUHE21/pDS56-derivatives allowing for IPTG-controlled expression of *clpB*, *clpB-K476C* and *clpG* were grown at 30 °C to early logarithmic phase (OD600: 0.2). Disaggregase expression was induced by IPTG addition (*clpB*, *clpB-K476C*: 25 μM IPTG, *clpG*: 100 μM IPTG). Production of disaggregases to similar levels was documented by SDS-PAGE followed by Coomassie staining. After 2 h, Luciferase activities were determined in a Lumat LB 9507 (Berthold Technologies). Here, 150 μL of cells were transferred into plastic tubes, 125 μL 0.25 mM Luciferin subsequently injected and luminescence determined for 10 s. Subsequently, 1 mL aliquots of cells were shifted to 42–50 °C for 15 min. Next, tetracycline (70 μg/mL) was added to the aliquots and cells were shifted to 30 °C. Luciferase activities were determined at 0, 30, 60, 120 and 180 min during the recovery phase. The activity of Luciferase prior to heat shock was set to 100%.

#### 2.5.2. Heat Resistance Assay

*E. coli* Δ*clpB* cells harboring the plasmid placIq and pUHE21-derivatives allowing for IPTG-controlled expression of *clpB*, *clpB-K476C* and *clpG* were grown at 30 °C to early logarithmic phase (OD600: 0.2). Disaggregase expression was induced by IPTG addition (*clpB*, *clpB-K476C*: 25 μM IPTG, *clpG*: 100 μM IPTG). Production of disaggregases to similar levels was documented by SDS-PAGE followed by Coomassie staining. After 2 h, 1 mL aliquots of cells were shifted to 50–53 °C for up to 120 min. Cellular viability (CFU) was determined before and after heat stress by spotting serial dilutions on LB plates and incubation for 24 h at 30 °C. Colony numbers before heat shock were set as 100% viability.

### 2.6. Cellular toxicity assay

*E. coli* cells harboring plasmid-encoded *clpB* or *clpG* alleles were grown in the absence of IPTG overnight at 30 °C. Serial dilutions were prepared, spotted on LB-plates containing different IPTG concentrations, and incubated for 24 h at indicated temperatures.

## 3. Results

### 3.1. ClpG Provides Efficient Protection Towards Severe Heat Stress

Expression of either *E. coli* or *P. aeruginosa* ClpG in *E. coli* K12 wild type cells leads to improved survival during severe heat stress [22,24]. The increased heat resistance can be explained by several scenarios. First, the co-existence of two disaggregation systems (ClpB and ClpG) increases the overall disaggregation capacity of cells. Second, ClpB and ClpG differ in substrate specificity and thereby together target an enlarged pool of aggregated proteins. Third, ClpB and ClpG differ in disaggregation activity and a higher potential of ClpG would provide increased cellular protection. In order to determine the individual contributions of the two disaggregases to stress resistance we separately expressed *E. coli clpB* and *P. aeruginosa clpG* (copy encoded on the genomic island) to similar levels in *E. coli ΔclpB* cells, which are highly sensitive to severe heat stress (Appendix A). *ΔclpB* cells harboring an empty vector control lost viability upon lethal heat shock to 50 °C or 53 °C within 30 min (Figure 1). Expression of *clpB* increased heat resistance, though prolonged incubation at high temperatures led to substantial loss of viability (e.g., 200-fold after 60 min incubation at 50 °C). In contrast, expression of *clpG* strongly protected *ΔclpB* cells at 50 °C and viability was only reduced six-fold after 120 min incubation (Figure 1). Although protection conferred by *clpG* at 53 °C was more transient, survival rates were still >20-fold higher as compared to *ΔclpB* cells expressing *clpB*. These findings suggest expression of ClpG is sufficient to confer enhanced heat protection to *E. coli* cells and does not necessarily require the presence of the ClpB disaggregase.

### 3.2. In Vitro ClpG is More Efficient in the Disaggregation of Diverse Aggregated Model Substrates

The different capacities of ClpB and ClpG in providing thermal protection might be caused by differences in disaggregation activities. We therefore analyzed the efficiencies of the two disaggregases towards various aggregated model substrates in vitro. Disaggregation activities were determined by directly monitoring the decrease in aggregate turbidity in light scattering experiments. We included the ClpB-K476C M-domain mutant, which has higher ATPase and disaggregation activity as compared to ClpB wild type due to dissociation of M-domains from the AAA-1 ring [16], as an additional reference (Figure 2a). All disaggregation reactions using ClpB or ClpB-K476C included the DnaK system, which is required for aggregate targeting and the activation of ClpB. The overall disaggregation rates differed substantially for the diverse model substrates ranging from 0.17 (Citrate synthase, ClpB) up to 3.38% Δturbidity/min (Malate Dehydrogenase, ClpG) (Figure 2b–e). When comparing the different disaggregation systems, we observed the following trends. First, ClpG was always more powerful as compared to ClpB and even outplayed the activated ClpB-K476C mutant. ClpG disaggregation rates were up to 6.2-fold (α-Glucosidase) higher as compared to ClpB. Second, ClpG exhibited robust disaggregation activity towards all model substrates including Citrate Synthase, which was hardly processed by ClpB or ClpB-K476C. Third, ClpB only exhibited high disaggregation activity (2.2-fold less than ClpG) towards aggregated malate dehydrogenase.

The differences in disaggregation activities between ClpB (low) and ClpG (high) must be based on specific features of protein aggregates formed by the individual model substrates. Protein aggregates that form in vivo are heterogenous and include a multiplicity of diverse protein species [1,2]. In order to generalize our findings obtained with single model substrates and to better mimic in vivo disaggregation, we generated heterogenous aggregates ex vivo by subjecting soluble lysates from *E. coli ΔclpB* cells to heat shock. Aggregates were subsequently isolated by centrifugation (Figure 3a). Next, we determined the disaggregation efficiencies of ClpB, activated ClpB-K476C and ClpG towards these complex aggregate mixtures in two ways. First, we ^3^H-labeled the lysate proteins using *N*-Succinimidyl [2,3-^3^H] propionate. This reagent reacts with exposed lysines residues and therefore labels bulk proteins. Disaggregation was monitored by determining the amount of disaggregated ^3^H-labeled proteins in the soluble fraction (Figure 3b,). Second, the lysate included Luciferase as reporter, allowing us to determine refolding efficiencies of aggregated Luciferase as alternative readout for disaggregation (Figure 3d,e).

ClpG showed a 2.2-fold higher disaggregation activity towards ^3^H-labeled aggregated bulk proteins as compared to ClpB and its disaggregation rate was comparable to ClpB-K476C (Figure 3b,c). The total disaggregation efficiencies (% solubilized ^3^H-labeled proteins) of the diverse disaggregation systems were largely similar after 120 min (ClpB: 57 ± 6%; ClpG: 64 ± 5%; ClpB-K476C: 67 ± 8%). One third of the generated ^3^H-labeled protein aggregates remained resistant towards disaggregation. The refolding yields of aggregated Luciferase were much lower, indicating heterogeneity among the formed protein aggregates with Luciferase-including aggregates being more difficult to be disentangled. ClpG was most efficient and refolded 12.9 ± 1.2% of aggregated Luciferase at a 8.5-fold higher disaggregation rate as compared to ClpB wt (3.2 ± 0.5% refolded Luciferase after 120 min) (Figure 3d,e). ClpB-K476C reactivated 10 ± 1% of aggregated Luciferase, though its Luciferase disaggregation rate was 2.6-fold lower as compared to ClpG. The differences in disaggregation activities between the diverse systems determined for these complex protein aggregates fit overall well with those determined for the single model substrates. ClpG represents a more potent disaggregase as compared to ClpB and its disaggregation rate and efficiency of aggregate processing is similar or higher as compared to activated ClpB-K476C.

### 3.3. ClpB and ClpG Reveal Temperature-Dependent Differences in Disaggregation Potential In Vivo

To test whether differences in disaggregation activities of ClpB and ClpG are also observed in vivo, we made use of *E. coli ΔclpB* mutant cells constitutively expressing Luciferase and expressing plasmid-encoded ClpB, ClpB-K476C and ClpG disaggregases to similar levels upon addition of IPTG (Isopropyl-β-d-thiogalactopyranosid) (Appendix A). This setup allowed us to determine the efficiencies of Luciferase reactivation after denaturation at different temperatures (42–50 °C) as a readout for cellular disaggregation activities. Of note, the 100% reference values for recovery efficiencies determined as Luciferase activity at 30 °C were largely similar for all cells. Protein synthesis was inhibited by addition of the antibiotic tetracycline immediately after application of the 15 min heat stress for two reasons (Figure 4a). First, this procedure enables cells to directly react to the stress condition by triggering a heat shock response. Second, newly synthesized proteins are particularly vulnerable to misfolding during heat shock, and thereby constitute a substantial source of stress-induced aggregates. We consider this protocol superior for comparing the potential of the disaggregases.

All disaggregases worked highly efficiently in the reactivation of Luciferase aggregates formed at 42 °C and 46 °C (Figure 4b–d). ClpB was most efficient towards aggregates formed at 42 °C and refolding activities even exceeded 100%. These values can be explained by (i) the ongoing synthesis of Luciferase during heat stress and (ii) the rescue of aggregated Luciferase already present at 30 °C. Increasing chaperone levels upon heat shock and the arrest of translation by tetracycline addition might reallocate chaperone resources towards disaggregation during the 30 °C recovery period [33]. In case of aggregates formed at 46 °C, we noticed that ClpG allowed for faster Luciferase recovery from the aggregates, though the total refolding yields were comparable for all disaggregases (Figure 4b). Substantial differences in disaggregase activities were noticed when cells were shifted to 50 °C. Here, ClpB only showed minor activity (5.7% refolded Luciferase after 120 min), while ClpG still showed robust activity (32% refolded Luciferase after 120 min) (Figure 4c). Accordingly, ClpB retained only 3.6% of its activity towards Luciferase aggregates formed at 50 °C as compared to those generated at 42 °C whereas ClpG retained 29% activity (Figure 4d). Expression of the activated K476C mutant of ClpB improved ClpB-dependent disaggregation, however, it stayed less efficient as compared to ClpG. To exclude that severe differences in cellular viabilities after exposure to 50 °C for 15 min are the basis for the substantial differences in Luciferase recovery, we re-determined heat resistance in the diverse strains (Appendix A). After 15 min incubation at 50 °C viability was reduced by 50% for *clpB* and *clpB-K476C* and 25% for *clpG* expressing *ΔclpB* cells. This two-fold difference is unlikely to explain the six-fold difference determined for Luciferase activation. Furthermore, ClpB and ClpB-K476C do not differ in providing heat resistance at 50 °C while showing different disaggregation activities. We therefore conclude that protein aggregates forming at 50 °C show remarkable differences in their susceptibility towards disaggregation by ClpB and ClpG.

### 3.4. ClpG but not ClpB Shows Robust Disaggregation Activity towards Luciferase Aggregates Formed at Increasing Temperatures

We speculated that the shift to increasing temperatures modulates the aggregation process, thereby alternating the nature of the formed aggregates in two ways. First, it will increase the concentration of misfolded proteins and thus overall protein aggregation. Second, it will cause more rapid and complete misfolding of proteins. This will likely increase the number of interactions between the aggregating protein species causing the formation of “tighter” aggregates. These differences in the state of the protein aggregates might in turn lead to differences in the disaggregation efficiencies by ClpB and ClpG. To test for both parameters, we used Luciferase as a model substrate and denatured low (25 nM) or moderate (200 nM) concentrations of the reporter at 42 °C to account for differences in the concentration of misfolded proteins. Additionally, we heat shocked 200 nM Luciferase at 46 °C or 53 °C to account for differences in the kinetics and degrees of protein misfolding. The diverse aggregated species were subjected to ClpB and ClpG-mediated protein disaggregation and refolding efficiencies were determined (Figure 5). Disaggregation of the aggregates by ClpB was highly sensitive to the aggregation conditions and thus the nature of Luciferase aggregates formed. ClpB/DnaK refolded 18.6% of Luciferase aggregates generated at 42 °C (25 nM Luciferase) and this efficiency dropped by 7.6-fold for aggregates formed at 53 °C (200 nM) (Figure 5). In stark contrast, ClpG disaggregation efficiencies were robust and hardly affected by the aggregation conditions (26.1–37.9% recovery, respectively). As a consequence, ClpG activity outperformed ClpB by 11.3-fold at the most extreme aggregation condition (200 nM Luciferase, 53 °C), but only 1.4-fold at the mildest condition (25 nM, 42 °C) (Figure 5b,c). ClpB-K476C exhibited higher disaggregation activities as compared to ClpB wild type, indicating that persistent activation of the AAA+ engine can, in part, compensate for the limitations of ClpB wild type. However, ClpB-K476C disaggregation activity still dropped by 2.3-fold for the most severe aggregation conditions (200 nM Luciferase, 53 °C) and Luciferase refolding efficiencies were 2-fold lower as compared to ClpG at this temperature (Figure 5b,c). The trend of these in vitro findings is overall similar to the observed Luciferase disaggregation in vivo: The ClpB disaggregase is becoming increasingly inefficient towards protein aggregates that form at very high temperatures, whereas ClpG retains substantial activity. This trend is more pronounced in the in vitro reconstituted system, eventually because in vivo the presence of chaperones (e.g., Hsp70, sHsps) attenuates or modifies protein aggregation.

### 3.5. Surface Properties of Luciferase Aggregates Do Not Explain Differing Disaggregation Activities

We next sought to determine the molecular basis for the differing disaggregation activities of ClpB and ClpG towards Luciferase aggregates formed in vitro at different substrate concentrations and denaturing temperatures. First, we determined the size of the aggregates by dynamic light scattering (DLS) (Figure 6a). DLS revealed an average particle radius of 157 nm for Luciferase aggregates generated at mildest denaturation conditions (25 nM, 42 °C). The particle radius was larger upon higher concentrations of 200 nM Luciferase (486 nm) and slightly increased to 524 nm and 553 nm upon denaturation at 46 °C and 53 °C, respectively. Both ClpB and ClpB-K476C disaggregation activities differed substantially for the diverse 200 nM Luciferase aggregates (Figure 5). Therefore, the minor differences in aggregate size determined here can hardly explain the strong impact of the denaturation temperatures on ClpB activity.

Next, we tested for differences in the surface properties of the Luciferase aggregates and monitored the binding of the fluorophore bis-ANS (Figure 6b). Bis-ANS binds to hydrophobic patches, causing an increase of its fluorescence. We observed a strong increase in bis-ANS fluorescence in the presence of all Luciferase aggregates and the maxima of fluorescence intensities were blue-shifted from 513 nm to 484 nm. Bis-ANS fluorescence was approx. 8-fold lower for aggregates formed during the mildest denaturation conditions and at the lowest concentration (25 nM, 42 °C) as compared to all other aggregates. The accessible surface of these aggregates is on average 2.5-fold lower, taking into account the determined particle sizes and assuming a spherical shape of the aggregates. This suggests that the surface hydrophobicity of these Luciferase aggregates formed at 42 °C with 25 nM is lower as compared to the other aggregates. Importantly, we did not detect differences in bis-ANS binding to the 200 nM Luciferase aggregates (Figure 6b, Appendix A), implying that the differing disaggregation activities are not caused by different surface properties.

Unaltered surface hydrophobicities of the 200 nM aggregates predict that the, in part, very low ClpB disaggregation activities are not caused by defects in aggregate binding. ClpB-mediated disaggregation is initiated by binding of its partner chaperone DnaK to the aggregate surface. DnaK and ClpG do not cooperate in protein disaggregation but rather compete for aggregate binding. This explains why addition of the DnaK system strongly inhibits ClpG-mediated protein disaggregation [22]. We therefore determined the inhibitory efficiencies of the DnaK chaperone system (KJE) towards ClpG-mediated disaggregation for the diverse Luciferase aggregates (Figure 6c). This represents an indirect measure for the degree of DnaK binding to the protein aggregates, which in turn determines the efficiency of ClpB recruitment. We determined a similar reduction of ClpG activities in presence of KJE for all Luciferase aggregates. The minor reactivation of Luciferase from 25 nM aggregates by the DnaK system is consistent with previous reports showing disaggregation activity of KJE towards small-sized aggregates [34]. The almost identical inhibitory efficiency of DnaK suggests that DnaK binds equally well to all Luciferase aggregates. To directly demonstrate binding competition between ClpG and DnaK, we incubated ClpG in absence or presence of DnaK with aggregated Luciferase. ClpG binding was monitored by determining the amount of ClpG in the aggregate-containing pellet fraction after centrifugation (Figure 6d). Here, we used the ATPase-deficient ClpG-E383A/E732A mutant harboring mutations in the Walker B motifs of both ATPase domains (ClpG-DWB). This ClpG mutant allows for stable binding to Luciferase aggregates in presence of ATP, causing an increased abundance of ClpG in the pellet fraction as compared to a control lacking Luciferase aggregates. The addition of DnaK reduced the binding of ClpG-DWB to aggregated Luciferase by 1.86-fold and the amount of the disaggregase in the pellet fraction was similar to the control reaction (Figure 6d). This demonstrates that KJE can outcompete ClpG for aggregate binding explaining inhibition of ClpG disaggregation activity. We confirmed the inhibitory effect of the DnaK system also for aggregated Malate Dehydrogenase and α-Glucosidase (Appendix A). Thus, the lower disaggregation activities of the ClpB/DnaK system towards these model substrates cannot be explained by less efficient aggregate binding.

### 3.6. ClpG Exhibits High Unfolding Power in Contrast to ClpB

The characterization of Luciferase aggregates largely excludes that aggregate sizes and surface properties are causative for the substantial differences in disaggregation activities of ClpB and ClpG. We speculated that the denaturation of Luciferase at higher temperatures (e.g., 53 °C) leads to a more rapid and global unfolding of Luciferase, resulting in an increased number of interactions between aggregating species. The resulting aggregates will be “tighter” and, consequently, a higher threading force will be required to break intermolecular interactions and extract misfolded protein species from the aggregate. To test for differences in threading and unfolding power between ClpB and ClpG we made use of the fusion construct Luciferase-YFP. Luciferase-YFP forms mixed aggregates composed of misfolded Luciferase and native YFP, which resist unfolding (Figure 7a). These aggregates are efficiently solubilized by ClpB, yet the fused YFP is not unfolded during the disaggregation reaction, indicating that ClpB has low unfolding power [21]. This low activity is caused by repressing M-domains as overruling activity control in ClpB-K476C allows for YFP unfolding [16,35]. Therefore, we followed ClpB and ClpG-mediated disaggregation of heat-aggregated Luciferase-YFP by turbidity measurements and simultaneously monitored YFP fluorescence (Figure 7b,c). This allowed us to directly correlate disaggregation and unfolding activities. Disaggregation of aggregated Luciferase-YFP by ClpB, ClpB-K476C or ClpG proceeded with similar kinetics and efficiencies (Figure 7b). ClpB shows higher disaggregation activity towards Luciferase-YFP as compared to aggregated Luciferase (Figure 2b). This can be attributed to the fused YFP, which likely modulates aggregate structure making it more amenable to ClpB activity. Similarly, Luciferase-YFP aggregates can be partially processed by the DnaK system alone, in contrast to Luciferase aggregates generated under the same denaturation conditions [21]. While all disaggregases worked efficiently on the Luciferase-YFP aggregates, strong differences between ClpB and ClpG were noticed when comparing the YFP unfolding activities. YFP fluorescence hardly dropped during ClpB-mediated disaggregation, demonstrating that YFP is resistant towards ClpB threading (Figure 7c). In contrast, activated ClpB-K476C was proficient in YFP unfolding as evident from the initial decrease of YFP fluorescence. The recovery of YFP fluorescence at later time points of the ClpB-K476C disaggregation reaction reflects refolding of the YFP moiety. Fastest and most pronounced loss of YFP fluorescence was observed during ClpG-mediated disaggregation and was confirmed by recording YFP spectra at the end of the disaggregation reactions (Figure 7, Appendix A). YFP unfolding by ClpG was not observed for native Luciferase-YFP (Appendix A). This demonstrates that YFP is only unfolded during an ongoing disaggregation reaction that starts with the ClpG-mediated threading of an unfolded Luciferase moiety. These findings demonstrate that ClpG applies high unfolding forces during disaggregation, distinguishing it from the ClpB disaggregase. This enhanced unfolding power explains its superior disaggregation activity.

## 4. Discussion

In this study we performed a thorough comparison of the disaggregation activities of the widespread ClpB/Hsp70 (DnaK) bi-chaperone disaggregation system and the standalone disaggregase ClpG as exemplified by *P. aeuruginosa* ClpG_GI_. Using a variety of substrates, we showed that ClpG is a more potent disaggregase, both in vitro and in vivo. ClpG shows robust disaggregation activity that is largely independent of the applied stress regimes triggering protein unfolding and aggregation. In contrast, ClpB acts poorly on protein aggregates that form during most severe heat stress. Differences between ClpB and ClpG activities cannot be explained by aggregate sizes or surface properties, as those hardly differ between aggregates that are rather well or poorly processed by ClpB. Accordingly, Hsp70 efficiently binds to protein aggregates that are nevertheless poorly processed by its partner ClpB.

We therefore suggest that elevation of the denaturation temperatures changes aggregate structure in a different way. Extreme temperatures will cause fast and global unfolding of substrates. This will likely increase the number of interactions between aggregating protein species and lead to the formation of protein aggregates that are “tighter”. This predicts that for solubilization of those aggregates, a particularly high threading force of the disaggregase is required in order to break the multiple interactions between the aggregated proteins. Indeed, we show that ClpG has a higher threading power as compared to ClpB, as it can unfold a stable YFP domain during a disaggregation reaction in contrast to ClpB (Figure 7). This mechanistic restriction of ClpB is caused by its regulatory M-domains, which engulf the AAA-1 ring when a substrate is bound in the central translocation channel [36]. The formation of this repressing belt involves head-to-tail interactions between adjacent M-domains of ClpB [19]. As a result, ClpB is only transiently activated by the Hsp70 partner and can only shortly apply high threading forces [37]. M-domains of ClpG are much shorter and lack one wing of the coiled-coil structure. Therefore, a ClpB-type mode of regulation by M-domains cannot be operative in ClpG, mechanistically explaining why ClpG has a stronger threading activity. Overruling M-domain repression in ClpB-K476C increases threading power, and accordingly leads to higher disaggregation activities that were, in part, similar to those of ClpG (Figure 2a, Figure 3b and Figure 5a). However, expression of ClpB-K476C is toxic in *E. coli* cells [16] (Appendix A), indicating that the ClpB disaggregase needs to be tightly regulated and cannot evolve into a more powerful disaggregase. We did not observe toxic effects when expressing *P. aeruginosa* ClpG in *E. coli* cells (Appendix A). This implies that ClpG and ClpB-K476C differ in substrate binding specificities, as both exhibit high threading power. We assume that ClpG binds with high specificity to protein aggregates but not other cellular targets, whereas ClpB-K476C binds and unfolds some proteins with crucial functions for cellular viability.

The increased threading power of ClpG is sufficient to explain why it provides stronger heat resistance to bacteria in comparison to ClpB. This raises the question, why is ClpB, but not ClpG, widespread in bacteria? When analyzing the disaggregation capacity of ClpB in vivo, we observed that the ClpB/DnaK system works very efficiently for elevated temperatures up to 46 °C (Figure 4). We assume that most bacterial cells in their natural environment do not usually face much higher temperatures. Furthermore, temperature upshifts in nature typically follow a gradient but are not as abrupt as applied in this study. A slower upshift will allow cells to trigger a heat shock response and produce more disaggregases (ClpB, DnaK/DnaJ), but also small heat shock proteins (sHsps), which modulate protein aggregation and facilitate protein disaggregation by DnaK/ClpB [38]. Such transcription-controlled mechanisms will increase disaggregation power and thereby the disaggregation activity of widespread ClpB will be sufficient to protect cells from heat stress under most circumstances. This might explain why only 2% of total *E. coli* genomes encode for the ClpG disaggregase. An abrupt and severe heat stress to 50 °C will likely affect cellular transcription and translation [39] and prevent cells from eliciting a robust heat shock response. This will limit the activity of ClpB/DnaK but not ClpG, which is constitutively expressed in stationary phase at a wide temperature range [22]. Other factors might also contribute to the reduction of ClpB/DnaK disaggregation activity at 50 °C, including different thermal stabilities of the disaggregase system involved (e.g., DnaK, DnaJ) and changes in ATP availability that might differently impact ClpB and ClpG-mediated disaggregation reactions. Our finding that only ClpG exhibits robust disaggregation activity in vivo upon heat shock to 50 °C predicts that severe stress conditions represent a selection pressure and increase the frequency of ClpG prevalence in a bacterial population. Indeed, ClpG prevalence is strongly increased up to 55% in *E. coli* isolates from raw milk cheese [24,28]. These cells face an abrupt upshift to extreme temperatures (57–68 °C) during the cheese production process and therefore have to rely on the more potent ClpG disaggregase for survival. Therefore human-induced temperature upshifts during food production and sterilization procedures select for bacteria equipped with more powerful disaggregases. As *clpG* is typically located on plasmids or mobile genomic islands, lateral transfer and spread between bacterial species can occur. Thus, in the future, current intervention methods might no longer suffice to reduce the number of contaminating bacteria to safe levels. It will be therefore important to analyze the mechanism of ClpG-mediated temperature tolerance, representing a key determinative factor of highly heat resistant cells. This might allow for the improvement of existing intervention protocols to ensure the efficient killing of *clpG*-encoding strains.

## 5. Conclusions

Severe heat stress causes massive and detrimental protein aggregation in cells. Highly heat resistant bacteria encode for the standalone disaggregase ClpG. We show that ClpG is more potent as compared to the widespread ClpB disaggregase. ClpG exerts higher unfolding forces, enabling it to process highly stable aggregates. Mechanistically, this explains how ClpG can provide superior heat resistance to bacterial cells.

## Figures and Tables

**Figure 1 biomolecules-09-00815-f001:**
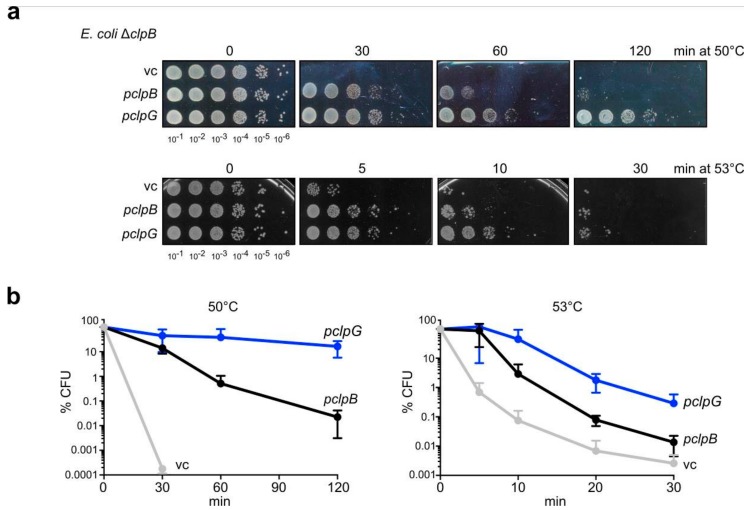
ClpG provides superior heat resistance. (**a**) *E. coli ΔclpB* cells harboring plasmids for expression of *E. coli clpB* or *P. aeruginosa clpG* (vc: empty vector control) were grown at 30 °C to mid-logarithmic growth phase and shifted to 50 °C or 53 °C. Serial dilutions of cells were prepared at the indicated time points, spotted on LB plates and incubated at 30 °C. (**b**) Colony numbers (CFU) were determined after 24 h and set to 100% for non-heat shocked samples. Standard deviations are provided (n = 3: 50 °C; n = 2: 53 °C).

**Figure 2 biomolecules-09-00815-f002:**
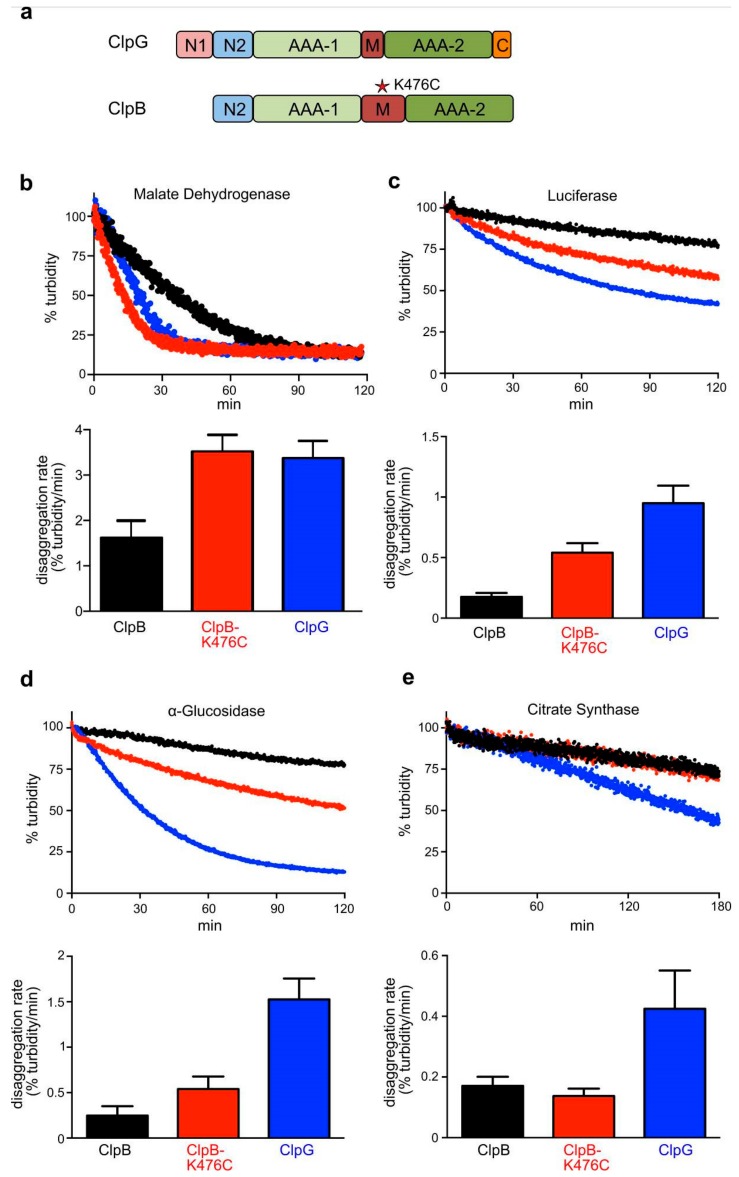
ClpG exhibits highest disaggregation activity towards diverse model substrates. (**a**) Domain organizations of ClpB and ClpG. Both Hsp100 proteins harbor two ATPase domains (AAA-1, AAA-2), a homologous N2 domain and a middle (M) domain. ClpG harbors additionally the N-terminal N1 domain and a C-terminal extension (C). The ClpB M-domain mutant K476C has increased ATPase and disaggregation activity due to dissociation of repressing M-domains. (**b**–**e**) Disaggregation of aggregated malate dehydrogenase (**b**), Luciferase (**c**), α-Glucosidase (**d**) and Citrate Synthase (**e**) by ClpB, ClpB-K476C or ClpG was monitored by light scattering. Sample turbidities at 0 min were set to 100%. Disaggregation reactions with ClpB and ClpB-K476C included the cooperating DnaK system. Disaggregation rates were determined based on the linear decrease in sample turbidity. Standard deviations are provided (n = 3).

**Figure 3 biomolecules-09-00815-f003:**
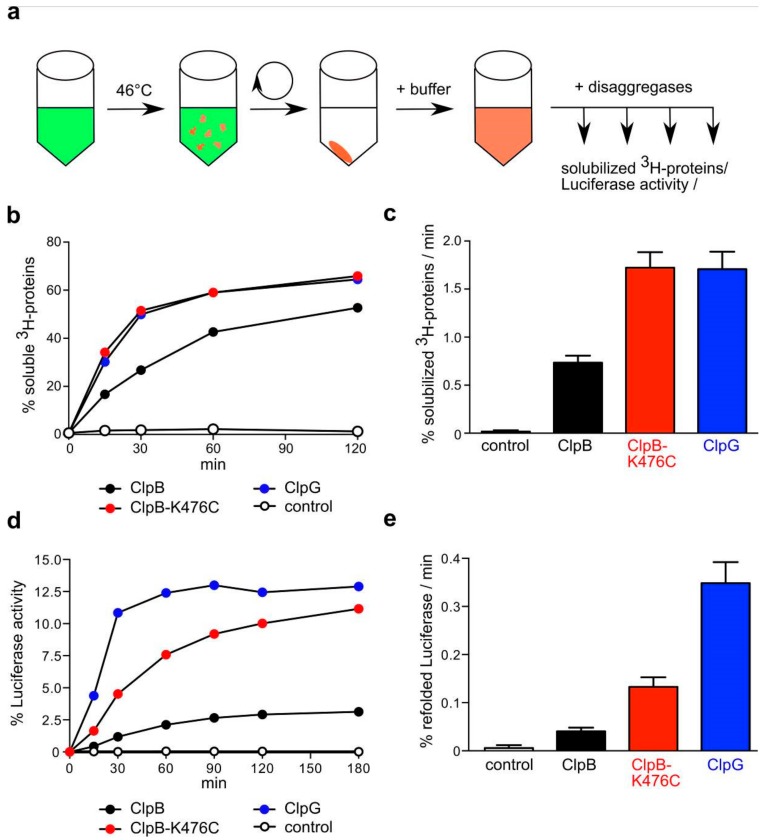
ClpG has higher disaggregation activity towards complex aggregates. (**a**) Soluble ^3^H-labeled proteins from an *E. coli ΔclpB* cell lysate (including Luciferase) were heat shocked to 46 °C for 15 min. Protein aggregates were isolated by centrifugation and resuspended in buffer. Disaggregase activities were monitored by determining the amount of solubilized ^3^H-labeled proteins (**b**,**c**) or Luciferase refolding (**d**,**e**). Disaggregation reactions with ClpB and ClpB-K476C included the cooperating DnaK system. Luciferase activities before heat shock were set to 100%. Rates of protein solubilization (**c**) and Luciferase reactivation (**e**) were determined. Standard deviations are given (n = 3).

**Figure 4 biomolecules-09-00815-f004:**
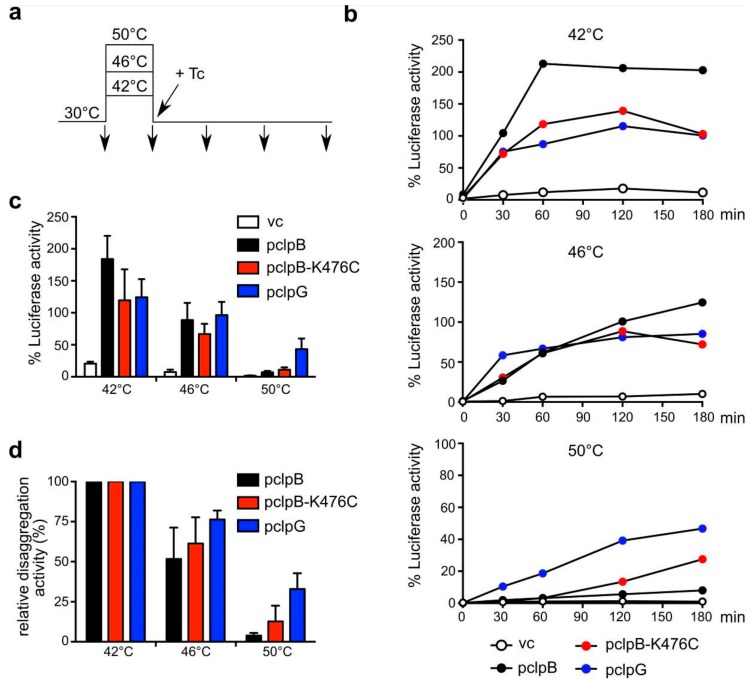
ClpG becomes a superior disaggregase in *E. coli* cells upon severe heat stress. (**a**,**b**) *E. coli ΔclpB* cells harboring plasmids for constitutive expression of Luciferase and IPTG-controlled expression of *E. coli clpB, E. coli clpB-K476C* or *P. aeruginosa clpG* (vc: empty vector control) were grown at 30 °C to mid-logarithmic growth phase and shifted to 42–50 °C for 15 min. Tetracycline was added directly after heat shock and cells were shifted to 30 °C. Luciferase activities were determined before and after heat stress and during the recovery period at 30 °C. Luciferase activities determined before heat stress were set as 100%. Absolute Luciferase refolding activities were compared after 120 min (**c**). Relative disaggregation activities (**d**) were calculated setting the absolute Luciferase activities determined in (**c**) after 42 °C heat shock as 100% for each disaggregase. Standard deviations are given (n = 3).

**Figure 5 biomolecules-09-00815-f005:**
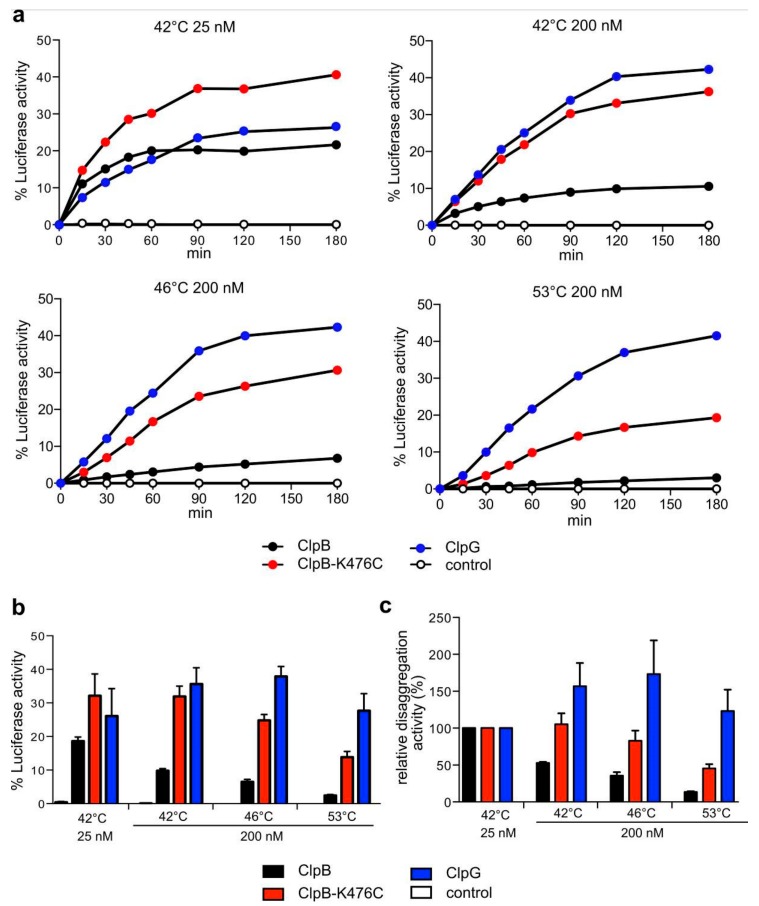
ClpG but not ClpB exhibits robust disaggregase activity. (**a**) Disaggregation activities of ClpB, ClpB-K476C and ClpG towards Luciferase aggregates generated at the indicated stress conditions were determined. Disaggregation reactions with ClpB and ClpB-K476C included the cooperating DnaK system. A reaction without chaperones served as control. Activities of native Luciferase (25 nM or 200 nM) were set as 100%. Absolute Luciferase refolding activities were compared after 120 min (**b**). Relative disaggregation activities (**c**) were calculated based on (**b**) by defining the activity of each disaggregase determined for 25 nM Luciferase aggregates generated at 42 °C as 100%. Standard deviations are given (n = 3).

**Figure 6 biomolecules-09-00815-f006:**
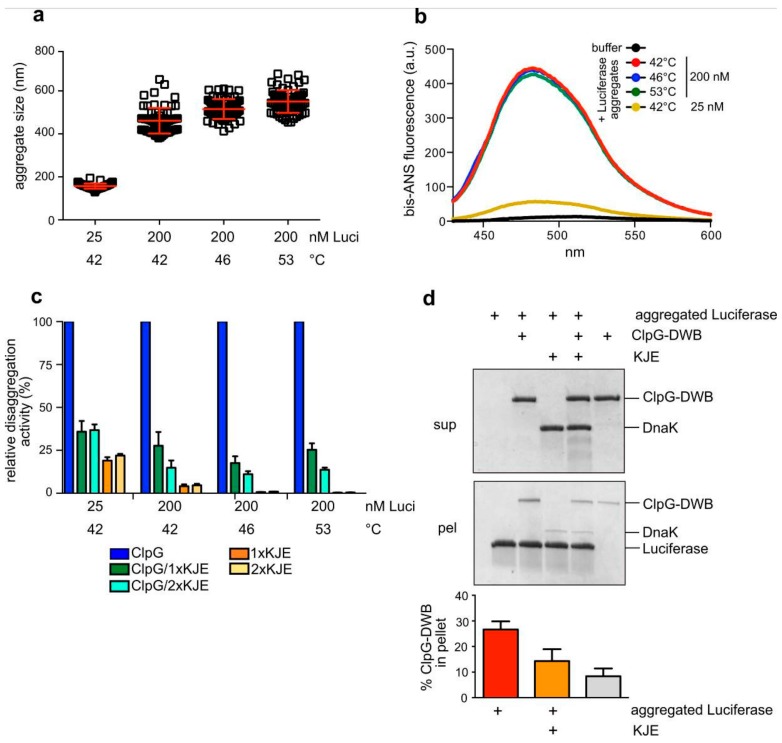
Characterization of Luciferase aggregates. (**a**) The sizes of Luciferase aggregates (radius: nm) were determined by dynamic light scattering. Denaturation conditions (temperature, Luciferase concentration (nM Luci)) applied to generate Luciferase aggregates are indicated. (**b**) Binding of bis-ANS to Luciferase aggregates. Denaturation conditions (temperature, Luciferase concentration) applied to generate Luciferase aggregates are indicated. The buffer control refers to bis-ANS fluorescence in the absence of Luciferase aggregates. (**c**) Disaggregation activities of ClpG and the DnaK system (KJE) towards Luciferase aggregates (Luci) generated at the indicated stress conditions were determined. The activities of ClpG determined in absence of the DnaK system were set as 100% for each specific Luciferase aggregate. Standard deviations are provided (n = 3). (**d**) ATPase-deficient ClpG-DWB was incubated in absence or presence of the DnaK system (KJE) with aggregated Luciferase. Soluble and insoluble fractions were separated by centrifugation and analyzed by SDS-PAGE and Coomassie staining. The amount of ClpG-DWB in the supernatant and pellet fractions (readout for aggregate binding) was quantified by Image J. A sample without Luciferase aggregates served as a control. Standard deviations are provided (n = 3).

**Figure 7 biomolecules-09-00815-f007:**
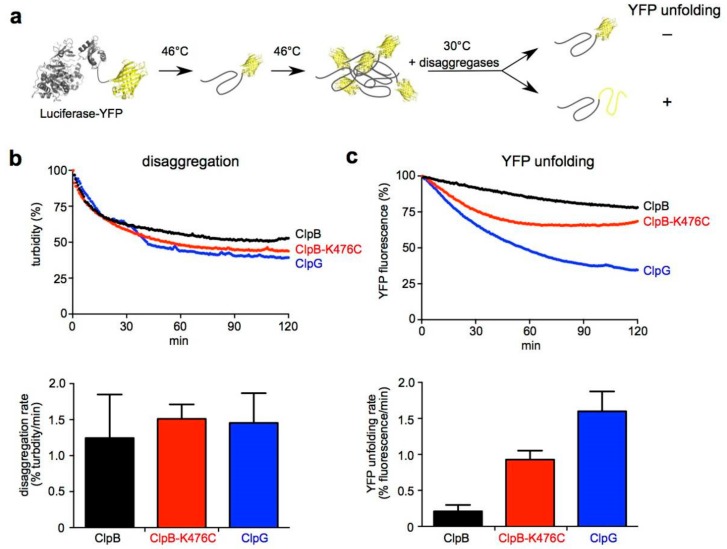
ClpG has high unfolding power. (**a**) Luciferase-YFP was incubated at 46°C leading to unfolding of the Luciferase moiety and the formation of mixed aggregates comprising misfolded Luciferase and native YFP. YFP fluorescence was monitored during disaggregation reactions as readout for unfolding power of the disaggregases. (**b**,**c**) Aggregated Luciferase-YFP was incubated with ClpB, ClpB-K476C or ClpG. Disaggregation reactions with ClpB and ClpB-K476C included the cooperating DnaK system. Disaggregation was monitored by light scattering (**b**). Sample turbidities at 0 min were set to 100%. Disaggregation rates were determined based on the linear decrease in sample turbidity. Changes in YFP fluorescence were simultaneously recorded (**c**). Initial YFP fluorescence was set as 100%. YFP unfolding rates were determined based on the linear decrease of YFP fluorescence. Standard deviations are provided (n = 3).

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
