# Peer review of "ClpG Provides Increased Heat Resistance by Acting as Superior Disaggregase"

_biomolecules, 2019, doi:10.3390/biom9120815_

Round 1

Reviewer 1 Report

In their manuscript, Katikaridis and colleagues have investigated the disaggregation efficiencies of the two major bacterial disaggregases ClpB and ClpG, using several protein aggregates both in vivo and in vitro. They first show that ClpG was more efficient than the generic ClpB chaperone in rescuing bacterial survival at high temperature following extended severe heat stress at 50°C in vivo. Further in vitro analysis with 4 different model substrate proteins and with native aggregates extracted form E. coli cells shows that ClpG has a more robust disaggregase activity than the DnaKJ-ClpB bichaperone. When Luciferase disaggregation and refolding were monitored in vivo in E. coli cells, the authors found that following a mild heat shock ClpB was a better in refolding luciferase than ClpG, while at higher and more stringent temperatures ClpG capacity was superior. Comparable results were obtained in vitro with purified luciferase. While both the size and surface properties of the aggregates were likely not responsible for such differences between the chaperones, the authors directly demonstrated that ClpG has a more robust substrate unfolding capacity than ClpB/DnaKJ, as judged by its increased ability to efficiently unfold stably folded substrates.

The manuscript is well written, the experiments presented are well controlled and the data are interesting and novel. The specific properties of the recently discovered ClpG disaggregase presented here show that a very powerful protein disaggregation at extreme temperatures of growth can be performed efficiently and without the need of a DnaKJ upstream chaperone. The observation that the protein unfolding ability of the chaperone is likely responsible is also remarkable.

Therefore, I strongly recommend this work and I have only minor comments, none of them being compulsory.

-The authors should indicate how the more stable binding of ClpG E383/E732A to substrate was tested (line 37 page 15).

-Although as proposed it very likely that the properties of the aggregates are not primarily responsible for ClpG powerful disaggregase activity, the presence of the small inclusion body proteins IpbA/B in protein aggregates could somehow impact ClpG capacities. Therefore, it would be interesting to discuss the possible impact of IbpA/B on ClpG superior capacity.

-The fact that DnaKJ does not contribute to ClpG activity at high temperature is remarkable. The authors should consider the possibility that the DnaKJ-dependence of ClpB might be in part responsible for the poor activity of ClpB at 50°C, a temperature that could impact DnaKJ itself.

- As stated in the text, the ClpB-K476C mutant has higher ATPase and disaggregation activity when compared to ClpB wild type. In almost if not all the experiments it performs at intermediate level, between ClpB and ClpG activities. Regarding the fact that ClpG has higher ATPase activity, it is very likely that ATP availability and hydrolysis at high temperature might play an important role in order to survive severe heat stress. Therefore, the authors should discuss this important point as well.

Author Response

Responses are shown in italics

-The authors should indicate how the more stable binding of ClpG E383/E732A to substrate was tested (line 37 page 15).

The experimental details of the binding experiment are described in the Materials and Methods section (2.4). We have modified the text to facilitate the understanding of the experimental procedure and the readout (page 5, lines 180-182; page 17, lines 454-458).

-Although as proposed it very likely that the properties of the aggregates are not primarily responsible for ClpG powerful disaggregase activity, the presence of the small inclusion body proteins IpbA/B in protein aggregates could somehow impact ClpG capacities. Therefore, it would be interesting to discuss the possible impact of IbpA/B on ClpG superior capacity.

The reviewer is correct, small heat shock proteins (like IbpA/B from E. coli) have been shown to modulate the disaggregation process. IbpA/B are almost non-detectable in non-stressed E. coli cells and are strongly upregulated upon heat shock. We speculate that the sHsps play a role at 42°C and 46°C heat shock temperatures helping particularly the DnaK/ClpB disaggregation system to reverse protein aggregation in agreement with earlier findings (e.g. Mogk et al., 2003, Molecular Microbiology 50:585-595). Most severe heat stress (50°C) likely affects cellular transcription and translation and therefore does not allow E. coli cells to efficiently elicit a heat shock response. This phenomenom has been described in several organisms including S. cerevisiae (Cherkasov et al., 2013, Current Biology 23:2452-62). In consequence the increase in levels of heat shock proteins (including sHsps) will be lower at 50°C as compared to less severe heat stress. This might also result in less efficient disaggregation by DnaK/ClpB. Since ClpG disaggregation activity is still robust at 50°C, it seems to be less dependent on sHsps. This is also consistent with our in vitro settings as all substrate proteins are aggregated in absence of sHsps. We have added these points to the discussion section (page 19, lines 557ff).

-The fact that DnaKJ does not contribute to ClpG activity at high temperature is remarkable. The authors should consider the possibility that the DnaKJ-dependence of ClpB might be in part responsible for the poor activity of ClpB at 50°C, a temperature that could impact DnaKJ itself.

The reviewer is correct, the DnaK and/or DnaJ chaperones might be partially thermolabile and loose activity at 50°C, thereby limiting ClpB-dependent protein disaggregation. We have discussed this possibility in a revised discussion section (page 19, lines 557ff).

- As stated in the text, the ClpB-K476C mutant has higher ATPase and disaggregation activity when compared to ClpB wild type. In almost if not all the experiments it performs at intermediate level, between ClpB and ClpG activities. Regarding the fact that ClpG has higher ATPase activity, it is very likely that ATP availability and hydrolysis at high temperature might play an important role in order to survive severe heat stress. Therefore, the authors should discuss this important point as well.

ClpG has a higher basal ATPase activity as compared to ClpB or ClpB-K476C. ATP hydrolysis by the activated ClpB-K476C mutant is strongly increased upon substrate binding (e.g. casein) and becomes higher as compared to the basal ClpG activity. We assume that ClpG ATPase activity is also stimulated by substrates but we could not yet establish appropriate conditions in vitro (such assay requires high, saturating concentrations of a soluble substrate protein; casein cannot be used as it does not stimulate ATP turnover by ClpG). This constraint currently prevents a direct comparison of ClpB and ClpG ATPase activities. We agree with the reviewer that ATP availability during severe heat stress might differently affect ClpB and ClpG function and added this aspect to the revised discussion (page 19, lines 557).

Reviewer 2 Report

This study by Katikardis et al. presents experiments designed to compare the activities of two bacterial disaggregases, namely the widespread ClpB chaperone that cooperates with the DnaK system in suppressing and reversing protein aggregation and the standalone ClpG protein. The authors showed that ClpG is more powerful dissagregase as compared to ClpB, especially under extreme heat stress. The authors suggested that ClpG disaggregase may protect bacteria against severe heat treatment. The manuscript is clear and concise. Likewise, the authors appear to have conducted their experiments carefully and with appropriate controls. However, there are some points that need addresing as presented below.

It would be worth describing the method used in this study to construct/produce of mutated variants of ClpB or ClpG, including ClpB-K476C. Was oligonucleotide-directed mutagenesis of clpB/clpG carried out in the study? Which of the luciferases was used in the study, eukaryotic or bacterial? A more accurate method of determining luciferase activity should be provided in the Materials and methods section. Have any substrates been used to determine an enzyme activity? The legend of Figure 2 is not consistent with its image. This figure should show (a) domain organizations of ClpB and ClpG disaggregases and results of dissagregation reactions of aggregated substrate proteins (b-e). Instead, Fig 2 shows that ClpG has higher disaggregase activity towards complex aggreagtes than ClpB wt. It looks like the right Fig 2 is missing and Fig 3 was also repeated in Fig 2.

Section Materials and Methods, point 2.6. Cellular toxicity assay should be moved to the Supplementary Material. I can not find in the main text of the manuscript an experiment using this method.

Lane 7, page 15: A control sample without Luciferase aggregates served as control.? Figure 6, panel d: How the percentage of ClpG-DWB in the pellet was estimated?

Lane 20, page 3: The reference and/or source for pDS56-derived expression vectors need to be provided. 

Figure 4: Relative disaggregation activities (d) were calculated based on (c) by defining the activity...

Figure 5: What is a control in this experiment?

Lines to be numbered continuously in the main text of the manuscript to facilitate the review process.

Author Response

Responses are shown in italics

- It would be worth describing the method used in this study to construct/produce of mutated variants of ClpB or ClpG, including ClpB-K476C. Was oligonucleotide-directed mutagenesis ofclpB/clpG carried out in the study? Which of the luciferases was used in the study, eukaryotic or bacterial? A more accurate method of determining luciferase activity should be provided in the Materials and methods section. Have any substrates been used to determine an enzyme activity? The legend of Figure 2 is not consistent with its image. This figure should show (a) domain organizations of ClpB and ClpG disaggregases and results of dissagregation reactions of aggregated substrate proteins (b-e). Instead, Fig 2 shows that ClpG has higher disaggregase activity towards complex aggreagtes than ClpB wt. It looks like the right Fig 2 is missing and Fig 3 was also repeated in Fig 2.

We have described the respective experimental procedures in more detail:

ClpB and ClpG mutant variants were generated by PCR-based oligonucleotide-directed mutagenesis. PCR products were incubated with DpnI, which only digests methylated DNA and transformed into E. coli XL1 blue cells afterwards. Firefly Luciferase was used throughout the study and we added a more detailed description on how its activity was determined in vitro and in vivo. We only determined the enzymatic activity of Firefly Luciferase. For all other substrates we monitored changes in the turbidity and solubility of aggregated proteins during the disaggregation reactions as most direct readout. We strongly apologize for the incorrect Figure 2, which has been replaced by the correct one.

- Section Materials and Methods, point 2.6. Cellular toxicity assay should be moved to the Supplementary Material. I can not find in the main text of the manuscript an experiment using this method.

We refer to this experiment in the discussion section of the main text (page 19, lines 543-546) and we therefore prefer to keep the experimental details in the main manuscript.

- Lane 7, page 15: A control sample without Luciferase aggregates served as control.? Figure 6, panel d: How the percentage of ClpG-DWB in the pellet was estimated?

The reviewer is correct, a reaction without aggregated Luciferase served as control to monitor background binding of ClpG to the reaction tubes. The percentage of ClpG-DWB in the pellet fraction was determined by quantifiying the intensities of ClpG-DWB bands in soluble (Is) and insoluble (Ii) fractions by Image J and calculating Ii/(Is+ Ii). We have specified this quantification in the materials and methods section.

- Lane 20, page 3: The reference and/or source for pDS56-derived expression vectors need to be provided. 

A reference for the pDS56 expression vector is now provided.

- Figure 4: Relative disaggregation activities (d) were calculated based on (c) by defining the activity...

We have changed the definition of relative disaggregation activities: Relative disaggregation activities (d) were calculated based on the absolute Luciferase activities determined in (c) after 42°C heat shock. The regained Luciferase activity was separately set as 100% for each disaggregase.

- Figure 5: What is a control in this experiment?

In the control reaction no chaperones were added to aggregated Luciferase. We have now defined the control n the respective figure legend.

- Lines to be numbered continuously in the main text of the manuscript to facilitate the review process.

We have changed the line numbering accordingly.